

# Break in precipitation – temperature scaling over India predominantly explained by cloud-driven cooling

Sarosh Alam Ghausi[1,2], Subimal Ghosh[3,4] and Axel Kleidon[1]

[1] Biospheric Theory and Modelling Group, Max Planck Institute for Biogeochemistry, Jena 07745, Germany.
[2] International Max Planck Research School for Global Biogeochemical Cycles (IMPRS – gBGC), Jena 07745, Germany
[3] Department of Civil Engineering, Indian Institute of Technology Bombay 400076, India
[4] Interdisciplinary Programme in Climate Studies, Indian Institute of Technology Bombay 400076, India

*Correspondence to*: Sarosh Alam Ghausi (sghausi@bgc-jena.mpg.de)

**Abstract.** Climate models predict an intensification of precipitation extremes as a result of a warmer and moister atmosphere at the rate of 7%/K. However, observations in tropical regions show contrastingly negative precipitation-temperature scaling at temperatures above 23° - 25°C. We use observations from India and show that this negative scaling can be explained by the radiative effects of clouds on surface temperatures. Cloud radiative cooling during precipitation events make observed temperatures co-vary with precipitation, with wetter periods and heavier precipitation having a stronger cooling effect. We remove this confounding effect of clouds from temperatures using a surface energy balance approach constrained by thermodynamics. We then find a diametric change in precipitation scaling with rates becoming positive and coming closer to the Clausius – Clapeyron scaling rate (7%/K). Our findings imply that the intensification of precipitation extremes with warmer temperatures expected with global warming is consistent with observations from tropical regions when the radiative effect of clouds on surface temperatures and the resulting covariation with precipitation is accounted for.



# 1 Introduction

Climate models and observed trends have shown precipitation extremes to increase at the global scale with anthropogenic global warming (Fischer et al., 2013; Westra et al., 2013; Donat et al., 2016). This increase is largely explained by the thermodynamic Clausius-Clapeyron (CC) equation, suggesting a ≈7%/K increase in atmospheric moisture holding capacity per degree rise in temperature ("CC rate") (Allen & Ingram, 2002). Extreme precipitation is expected to increase at a similar rate (Trenberth et al., 2003; Held & Soden., 2006; O'Gorman & Schneider, 2009), as also shown by convection-permitting climate model projections (Kendon et al., 2014; Ban et al., 2015). Precipitation – temperature scaling rates, estimated using statistical methods and observed records, are widely used as an indicator to constrain this response (Lenderink et al., 2008; Wasko et al, 2014).

However, observed scaling rates show large heterogeneity globally, with significant deviations from the CC rate (Westra et al., 2014; Schroeer & Kirchengast, 2018). Deviations are larger in the tropical regions where scaling rates are mostly negative and precipitation extremes largely show a monotonic decrease or a sudden drop (hook) in scaling at high temperatures (Utsumi et al., 2011). These deviations have been studied and attributed to number of factors. Two primarily argued reasons include the moisture availability limitation at high temperatures (Hardwick et al., 2010) and dependence of scaling estimates on the wet event duration (Gao et al., 2018; Ghausi & Ghosh 2020; Visser et al., 2021). Cooling effects of rainfall events have also questioned the use of surface air temperature as scaling variable (Bao et al., 2017). Other scaling variables like atmospheric air temperature (Golroudbary et al., 2019), sampling temperatures before the start of storm (Visser et al., 2020), using measures of atmospheric moisture like dew point temperature (Bui et al., 2019) and integrated water vapor (Roderick et al., 2019) have been suggested as an alternative to surface air temperatures. The use of atmospheric moisture as a scaling variable has been criticized because it provides less insight about precipitation sensitivity to temperature and is also not entirely immune to cooling effects of rainfall (Bao et al., 2018). Other factors that can cause deviations in scaling includes the change in rainfall type from stratiform to convective (Berg et al., 2013; Molnar et al., 2015) and seasonality in precipitation (Sun et al., 2020). Owing to these uncertainties,



the use of scaling relationships derived from observations to infer future changes in extreme precipitation
in these regions remains debatable.

Here we show that a large part of uncertainties in this response and negative scaling rates in the tropics
are mainly caused by the radiative effect of clouds on surface temperatures. Precipitation events are
accompanied by strong cloud cover, which reduces the solar absorption at the surface and hence lowers
surface temperatures. This radiative cooling associated with precipitation can be significant in the tropical
regions where insolation and temperatures are high. As a result, regions and periods of more intense
precipitation cool more, and this affects their position in the scaling curve. This implies that temperature
observations are not independent of precipitation and this dependency obscures their scaling relationship.
Here we used a thermodynamic systems approach to remove this cooling effect from surface
temperatures. We then show that when this effect is being removed, no breakdown in the scaling
relationship is seen in observations and extreme precipitation then scales positively with temperature
close to CC rate.

To remove the effects of clouds, we used a surface energy balance formulation in conjunction with the
first and second law of thermodynamics (Kleidon & Renner, 2013). This approach provides us with
additional thermodynamic constraints on the turbulent flux exchange between surface and atmosphere.
We used this thermodynamically constrained model and force it with the "all-sky" and "clear-sky"
radiative fluxes. These fluxes are a standard product in NASA-CERES radiation datasets such that "all-
sky" fluxes are representative of observed conditions including the cloud effects while "clear-sky" fluxes
are diagnosed by removing the effect of clouds from the radiative transfer. Compounding the
thermodynamic constraint on turbulent fluxes together with the radiative fluxes helps us to estimate "all-
sky" and "clear-sky" temperatures that includes and excludes the radiative effects of clouds respectively.

We then evaluate this effect and its impact on precipitation-temperature scaling using observations from
India. India is a tropical country where the extreme precipitation and the resulting floods have intensified
over the past years (Goswami et al., 2006) and are expected to increase in the future (Katzenberger et al.,



2021). However, extreme precipitation–temperature scaling is largely negative over most of India (Vittal
et al., 2016; Sharma et al., 2019), which is in contrast to the observed trends (Roxy et al., 2017). We here
attempt to resolve this inconsistency in precipitation – temperature scaling by removing the cloud cooling
effects from surface temperatures. To do this, we use gridded precipitation – temperature datasets that
were used in previous studies (Vittal et al., 2016; Mukherjee et al., 2018; Sharma et al., 2019; Ghausi et
al., 2020) and supplement it with the gridded radiative flux datasets to remove the cloud radiative effects.
More details on our surface energy-balance model and estimation of surface temperatures "with" and
"without" clouds are followed in the section 2.1 with the details of datasets being used in section 2.2. We
used these reconstructed temperatures to study the effect of clouds on precipitation – temperature scaling
over India. To estimate the precipitation – temperature scaling rates, we used the widely adopted statistical
methods. Details of them are further provided in section 2.3. Results are then presented and discussed in
section 3.

## 2 Methods and Data

### 2.1 Thermodynamically constrained energy balance model

To infer surface temperatures from the radiative forcing and remove the effects of clouds, we start with a
simple physical formulation of the surface energy balance. The surface of the Earth is heated by solar
absorption and downwelling longwave radiation. This heat is released back to the atmosphere through
surface emission of longwave radiation and exchange of turbulent fluxes of sensible and latent heat. This
balance between the incoming and outgoing energy fluxes at the Earth's surface is described by the
equation (1).

$$R_s + R_{ld} = R_{l,up} + J \qquad (1)$$

Here $R_s$ is the surface net solar absorption, $R_{ld}$ is the downwelling longwave radiation, $R_{l,up}$ is the
upwelling longwave radiation emitted from the surface and $J$ is turbulent flux exchange between surface
and the atmosphere (comprising of sensible and latent heat).  We neglect the ground heat flux, as it is
generally small when averaged over a few days or longer. While $R_s$ and $R_{ld}$ can be obtained using radiation
datasets for different sky conditions, the partitioning between $R_{l,up}$ and $J$ is poorly constrained by surface
energy balance alone. To have these additional constraints on $J$, we used a thermodynamic systems





approach to view the earth system. Similar approach had also been used in (Kleidon & Renner, 2013;
Kleidon et al., 2014; Dhara et al., 2016) and were found to very well capture the observed surface
temperatures, energy partitioning and climate sensitivities.
To do this, we conceptualize the surface atmosphere system as a heat engine, with warm Earth surface as
the heat source and cooler atmosphere being the sink (Figure 1). Heat and mass are transported within
this engine by the exchange of turbulent fluxes (J) between the surface and the atmosphere. The
differential radiative heating and cooling between the surface and the atmosphere maintains the
temperature difference and drives the vertical convective motion. The power (G) associated with the work
done by the heat engine required to sustain convective motion in form of vertical mixing and exchange
of turbulent fluxes can be derived simply using the first and second law of thermodynamics and can be
represented by the well-established Carnot limit as

$$G = J \left( 1 - \frac{T_a}{T_s} \right).$$
(2)

Detailed derivation about the same can be found in (Kleidon & Renner, 2013; Kleidon et al., 2014). Here
$T_a$ and $T_s$ are the representative temperatures of cold atmosphere and the hot surface respectively.
Both temperatures are inferred from their respective energy balances. The atmospheric temperature ($T_a$)
is assumed to be equal to the radiative temperature of atmosphere and is estimated using the outgoing
longwave radiation at top of atmosphere ($R_{l,toa}$)

$$T_a = \left( \frac{R_{l,toa}}{\sigma} \right)^{1/4}.$$
(3)

Here, $\sigma$ is the Stefan Boltzmann constant ($\sigma = 5.67 \times 10^{-8}$ Wm$^{-2}$K$^{-4}$). A correction of 15K was applied to
the radiative temperature to account for the assumption of black atmosphere and effective height of
convection (Dhara et al., 2016). We consider the atmosphere as opaque to terrestrial radiation and hence
it is assumed that all outgoing longwave radiation emitted into space originates from the atmosphere.
The heat engine source temperature i.e. surface temperature ($T_s$) can be expressed from the emitted
longwave radiation from the surface ($R_{l,up}$) as

$$T_s = \left( \frac{R_{l,up}}{\sigma} \right)^{1/4}.$$
(4)





Using the surface energy balance (Eq. 1), we can then express the surface temperature in terms of net
solar absorption, downwelling longwave radiation and turbulent fluxes (J) as
$$T_s = \left(\frac{R_s + R_{ld} - J}{\sigma}\right)^{1/4} . \qquad (5)$$

The differential radiative heating and cooling between the surface and the atmosphere maintains the
temperature difference and drives the vertical convective motion. Thermodynamics sets a limit to this
conversion and thus constrains the amount of turbulent flux exchange. Less turbulent fluxes result in a
hotter surface (Eq. 5), which will increase the temperature difference between the surface and atmosphere.
This will subsequently increase the efficiency term in the generation rate, the second term on the right-
hand side of Eq. (2). On the other hand, an increase in turbulent fluxes (J) increases the first term in the
generation rate of Eq. (2), but it will, in turn, reduce the surface temperature and temperature difference
between surface and atmosphere (Eq. 5). Thus, there exists a trade-off that sets the limit for the power to
maintain vertical energy and mass exchange between surface and the atmosphere. This limit is termed as
the maximum power limit and provides an additional constraint to surface energy balance partitioning
that we used here to infer surface temperatures.
Using Equations. (2), (3) and (5), the rate of work done (power) produced by the heat engine can be
expressed as a function of turbulent fluxes (J) as
$$G = J\left(1 - T_a \left(\frac{R_s + R_{ld} - J}{\sigma}\right)^{-1/4}\right). \qquad (6)$$

Note that power G = 0 when J = 0 or when $J = R_s + R_{ld} - R_{l,toa}$. Hence, there is a maximum $G_{max} = G$
$(J_{maxpower})$ for a value between $0 < J_{maxpower} < R_s + R_{ld} - R_{l,toa}$ . The optimum J that maximizes power was
calculated numerically. This flux was then used to determine the surface temperatures.
$$T_{s,maxpower} = \left(\frac{R_s + R_{ld} - J_{maxpower}}{\sigma}\right)^{1/4} \qquad (7)$$

Surface temperatures were estimated using Eq. 7 for "all-sky" and "clear-sky" radiative conditions using
radiative forcing from the NASA – CERES datasets. We then refer to these two temperatures derived
using Eq. 7 as "all-sky" and "clear-sky" temperatures.




## 2.2 Datasets used

Radiative fluxes of shortwave and longwave radiation at surface and top of atmosphere (TOA) were obtained from the NASA - CERES (EBAF 4.1) dataset (Loeb et al., 2018; Kato et al., 2018) and NASA CERES Syn1deg dataset (Doelling et al., 2013,2016). These datasets are available for both "all-sky" as well as "clear-sky" conditions at monthly and daily scale respectively with a 1° latitude x 1° longitude spatial grid resolution and were used as a forcing in our energy balance model. We evaluated our model using observations derived gridded temperature data from Indian Meteorological Department (IMD, Rajeevan et al., 2008). To estimate the precipitation – temperature scaling, we used daily gridded precipitation and temperature datasets with a spatial resolution of 1° latitude x 1° longitude from the Indian Meteorological Department (IMD, Rajeevan et al., 2008) and 3 hourly gridded rainfall data from NASA-TRMM_3B42 with a spatial resolution of 0.25° x 0.25°. We repeated the analysis using daily gridded precipitation and temperature data from the APHRODITE (Asian Precipitation Highly Resolved Observational Data Integration towards Evaluation) dataset, available at a spatial resolution of 0.25° x 0.25° (Yatagai et al., 2012). To further ensure robustness of our results, we also used 3 station-based daily precipitation – temperature observations in India (Mumbai Airport, Bangalore Airport and Chennai Airport) from global surface summary of the day (GSOD) data provided by National Oceanic and Atmospheric Administration (NOAA). Daily dew point temperatures were obtained from the ERA-5 reanalysis. Based on the availability of all datasets, the period of analysis was chosen from the years 2003 to 2015.

## 2.3 Estimation of precipitation – temperature scaling rates

Extreme precipitation events were scaled with observed, "all-sky" and "clear-sky" temperatures using two widely adopted scaling approaches: The Binning Method (Lenderink et al., 2008) and Quantile Regression (Wasko et al., 2014). For the binning method, we defined extreme precipitation events using a threshold of 99th percentile precipitation contained at each grid cell. Precipitation – temperature pairs were then divided into the increasing order of non-overlapping bins of 2 K width. Only those bins which have at least 150 data points have been considered for the analysis (Utsumi et al., 2011). The median value of each bin was then used to examine the variation of precipitation extremes with temperature. Bins


with temperature less than 3°C were discarded to remove the effects of freezing, thawing and snowfall.
To ensure that our results are not biased with the number of data points in each bin and bin sizes (which
may affect the nature of the scaling relationship), we further used the Quantile Regression method to
estimate the scaling rates.
Quantile regression estimates the conditional quantile of the dependent variable (in our case,
precipitation) over the given values of the independent variable (temperature). We first fitted a quantile
regression model between the logarithmic precipitation and temperature values at the target quantile of

95   99%

$$Log(P_i) = \beta_o^{99} + \beta_1^{99}(T_i) \; . \quad\quad (8)$$

Here $P_i$ denotes the mean daily precipitation intensity and $T_i$ is the daily mean temperature, and $\beta_o^{99}$ and
$\beta_1^{99}$ are the regression coefficients for the 99th quantile of precipitation. The slope coefficient $\beta_1^{99}$ is then
exponentially transformed to estimate the scaling rate ($\alpha_1$).

$$\alpha_1 = 100 \cdot \left(e^{\beta_1^{99}} - 1\right) \quad\quad (9)$$

The following methodology had been widely adopted to estimate the extreme precipitation – temperature
scaling in previous studies (Lenderink et al., 2008, 2010; Utsumi et al., 2011; Wasko et al., 2014; Schroeer
et al., 2018).

## 3 Results and Discussion

In this section, we first start by a quick evaluation of our thermodynamic approach by comparing the
estimated "all-sky" temperatures against observations. We then quantify the cloud radiative effects on
surface temperatures and check for its spatial consistency across regions. We then estimated precipitation
– temperature scaling rates by including and excluding the effect of clouds on surface temperatures. We
also used dew point temperature (a proxy measure for atmospheric moisture) as a scaling variable. Later,
we discuss our interpretation of scaling by excluding cloud effects from temperatures, its comparison with
the dew point scaling and its implications across regions.



### 3.1: Evaluating the modelled temperatures

"All-sky" temperatures were estimated using the daily observed radiative fluxes from CERES in conjunction with surface energy partitioning constrained by maximum power (see Equation 7). We found an extremely good agreement of these estimated temperatures when compared to surface temperature observations over India with $R^2 > 0.9$ and RMSE < 1.5 K over most regions (Figure 2). This signifies that our formulation strongly captures the surface temperature variation over India and thus validates our approach. We then extend this for clear-sky conditions by forcing our model with "clear-sky" radiative fluxes from CERES and estimating "clear-sky" temperatures. It is to note that "clear-sky" temperatures are reconstructed temperatures estimated by removing the effect of clouds from radiative transfer.

### 3.2: Estimating the cloud radiative cooling

We used the difference between the "all-sky" and "clear-sky" temperatures as a measure to quantify the effect of cloud-driven cooling during rainfall events. This cooling increases strongly with precipitation across regions, resulting in a stronger reduction in surface temperature with greater precipitation (Figure 3a). This cooling is predominantly caused by the substantial reduction in absorbed solar radiation at the surface for "all-sky" conditions compared to "clear-sky" conditions (Figure 3b). On the other hand, changes in longwave radiation are comparatively small and largely remain insensitive to precipitation.

To examine the spatial consistency in precipitation variability and associated cooling, we isolated extreme daily precipitation days over each grid. Figure 4a shows the mean magnitude of daily extreme precipitation events over India. Figure 4b shows the cloud-cooling associated with these days. This cooling effect of clouds and precipitation shows a clear, systematic variation across India. The cooling effect is greater where precipitation rates are high. In contrast, in the more arid regions in the northwest of India, the cooling effect almost disappears with low precipitation rates. Figure 4c further shows the mean "all-sky" temperature during these days. We find that the heaviest events occur at a relatively lower temperature as a result of stronger cooling. Figure 4d shows the mean number of rainfall days per year. More rainy days implies more cloudy conditions and thus a stronger cloud radiative cooling over that region. Having quantified this effect of cloud radiative cooling and its systematic variation across regions, we then estimate its impact on the precipitation – temperature scaling.





## 3.3 Impact on precipitation-temperature scaling


We performed a binning analysis (Lenderink et al., 2008) to understand the scaling of precipitation
extremes with temperature using observed temperatures as well as our estimated "clear-sky" and "all-sky"
temperatures. Precipitation events were isolated and binned into P-T pairs and the resulting scaling
relationships are shown in Figure 5. The scaling relationship using observed and "all-sky" temperatures
showed similar scaling behaviour (yellow and red lines in Figure 5a). Extreme precipitation increases
close to the CC rate up to a threshold of around 23° - 24°C, above which the scaling becomes negative.
This break in scaling behaviour with observed temperatures is consistent with the findings of previous
studies (Hardwick et al., 2010; Ghausi & Ghosh, 2020) and is commonly referred in literature as "hook"
or "peak structure" (Wang et al., 2017; Gao et al., 2018). However, when precipitation extremes are scaled
with "clear-sky" temperatures that excludes the cloud-cooling effect, the resulting scaling relationship
does not show a breakdown and increases consistently, close to the CC rate over the whole temperature
range (blue line in Fig. 5a). Similar results were obtained when the scaling curves were reproduced for
station-based observations (See Appendix A).
Previous studies (Hardwick et al., 2010; Chan et al., 2015; Wang et al., 2017) have attributed the break
in precipitation-temperature scaling to a lack of moisture availability as relative humidity tends to
decrease at high temperatures. To account for this effect of moisture limitation, some studies used dew
point temperature, a measure of atmospheric humidity, as an alternative scaling variable (Wasko et al.,
2018; Barbero et al., 2018). They showed that the breakdown and negative scaling disappear when scaled
with dew point temperatures (Zhang et al., 2019; Ali et al., 2021). To evaluate this interpretation and
compare it to ours, we used the dew point temperature from the ERA-5 reanalysis. We derived the extreme
precipitation scaling using this temperature (Figure 5b) and compared it to our "all-sky" and "clear-sky"
temperatures (Figure 5c).
At first sight, the scaling relationship using dew point temperatures looks very similar to our "clear-sky"
relationship (compare Figures 5a and 5b, but note the difference in temperature scale). Yet, its
interpretation differs because using dew point temperatures merely implies that the intensity of extreme
precipitation events scales with the moisture content of the air, with moister air resulting in higher
intensity events. Dew point scaling thus carries less insight about the response of extreme precipitation to




climate warming (Bao et al., 2018). To infer the precipitation sensitivity with temperature from dew point
scaling, one then needs to see how dew point temperatures change with actual temperatures ($dT_{dew}/dT$)
(Figure 5c). This is further demonstrated using equation 10.
$$\frac{dP}{dT} = \frac{dP}{dT_{dew}} \times \frac{dT_{dew}}{dT} \qquad\qquad \textbf{(10)}$$

If relative humidity remains unchanged, we would expect the dew point temperature to increase
continuously with surface temperature, representing a moisture increase of 7%/K. However, when dew
point temperatures are compared to "all-sky" temperatures (red line, Figure 5c), we note that a break
occurs in this scaling as well. Dew point temperatures increase with "all-sky" temperatures for colder
temperatures more strongly than what would be expected from an unchanged relative humidity when air
gets warmer. However, at temperatures of above 23° - 25°C, dew point temperatures fall, reflecting a
decrease in relative humidity that is typical for warm, arid regions. Thus, one does not see a breakdown
in precipitation - dew point scaling because the information on the breakdown is contained in how dew
point temperatures change with surface air temperatures (second term in equation 10). Similar findings
were also reported in Roderick et al (2019).
The scaling of dew point temperatures with "clear-sky" temperatures is much more uniform and consistent
across the whole temperature range and does not show a breakdown or a super CC scaling in the
relationship. This is because the "clear-sky" temperatures reflect the radiative conditions, and not the
effects of atmospheric humidity or clouds. In contrast, observed temperatures and "all-sky" temperatures
co-vary with cloud effects, which in turn are linked to precipitation and humidity, thus resulting in less
clear scaling relationships that are less straightforward to interpret. This further implies that moisture
loading of the atmosphere primarily occurs during the non-precipitating periods that are more
representative of clear-sky radiative conditions.
The breakdown in scaling effect can thus be explained by the cooler temperatures associated with
precipitation events. This cooling shifts the precipitation extremes to lower temperature bins while the
high-temperature bins then correspond to more arid regions or to the drier pre-monsoon season
temperatures with lower values of precipitation extremes. We refer to this as a "bin-shifting" effect. The
cooling effect is proportional to the amount of precipitation (Fig. 3A) and hence, the heavier the





precipitation, the stronger the cooling and bin shifting becomes. When the cloud cooling effect is
removed, as in the case of "clear-sky" temperatures, extreme precipitation then shows a scaling that is
consistent with the CC rate. This bin shifting effect arising due to the presence of clouds also causes a
decrease in relative humidity at higher temperatures.  This effect can be seen by the stronger increase in
dewpoint temperatures below 25°C, and the decline above this temperature (Figure 5c). The breakdown
in scaling is thus not directly related to changes in aridity or moisture availability, but rather to the
radiative effect of clouds on surface temperature.
To demonstrate the implications of our interpretation for precipitation scaling across regions, we
estimated regression slopes of 99th percentile precipitation events for both sub-daily (TRMM) and daily
(IMD & APHRODITE) precipitation with the different temperatures using the Quantile Regression
method (Wasko et al., 2014). We found that extreme precipitation scaling was negative for both, observed
and "all-sky" temperatures over most regions (Figure 6) except for the Himalayan foothills in the North
of India. The scaling rates for sub-daily extremes were slightly higher than those estimated for daily
extremes but yet remains negative over most grids. When the cooling effect of clouds is removed by using
"clear-sky" temperatures, extreme precipitation scaling then shows a diametric change and scaling
estimates come close to CC rates over most of the regions. A similar diametric change in the scaling was
also obtained with the APHRODITE precipitation dataset (Appendix B).
We note that negative scaling was also found over few regions of South-central and south-east India with
"clear-sky" temperatures at both daily and sub-daily scales (Figure 6 c,f). To our understanding, this
negative scaling is largely due to the cyclonic activities originating from Bay of Bengal during winter
months and resulting in heavy rains over these regions. These cyclonic systems thus cause very high
rainfall at very low temperatures which causes negative scaling. More work is needed to be done to resolve
these systems in conventional scaling approach and remains an important area for future research.
The effect of seasonality on precipitation scaling was also checked by producing the scaling curves for
different seasonal subsets (summer and winter monsoon). Our findings indicate that seasonality does have
an effect on observed scaling while the "clear-sky" scaling rates remains positive irrespective of the
seasons (see Appendix C).





The confounding effect between precipitation and temperature on observed scaling relationships
"apparent scaling" had also been argued by some recent studies (Bao et al. 2017; Visser et al., 2020). Our
results agree with these studies that the observed scaling relationships also reflect the impact of synoptic
conditions and cooling associated with precipitation events on temperature. However, we suggest that this
confounding effect is largely associated with cloud radiative effect, which is removed by our use of "clear-
sky" temperatures as a scaling variable. We also address the arguments raised to resolve apparent scaling
using dew point temperature (Barbero et al., 2018). Our results confirm that precipitation extremes scale
well with dew point temperatures as a measure for atmospheric moisture, but that the break in scaling
actually originates from the scaling of dew point temperatures with observed temperatures. This response
of dew point temperature to warming is further affected by the presence of clouds and associated radiative
cooling. "Clear-sky" temperatures are independent of the co-variations arising from cloud effects and are
thus a better, more independent measure and scaling variable to understand the precipitation response to
climate warming.

## 4 Summary and Conclusions

We showed that the observed negative scaling of extreme precipitation in India arises mostly from the
cloud radiative cooling of surface temperatures. When this effect is removed, we get a positive scaling
consistent with the CC rate. Scaling rates estimated from observed temperatures are thus likely to
misrepresent the response of extreme precipitation to global warming, because the cooling effects of
clouds make precipitation and temperature covary with each other. When this effect is removed by
estimating surface temperatures for "clear-sky" conditions, the scaling relationships with moisture content
and precipitation become much clearer and confirm the CC scaling of extreme precipitation events with
warmer temperatures. This explains the apparent discrepancy between the observed negative scaling rates
over India and the projected increase in precipitation extremes by climate models.
It is also important to note that the goal of our study was not to compare the accuracy of scaling estimates
from different gridded and station-based datasets, but rather to identify and remove the physical effects
that causes uncertainties in this response. Our methodology to remove the cooling effect of clouds from
surface temperatures significantly improves the scaling estimate for daily precipitation scaling.



While our study was confined over the Indian region, we would expect that cloud effects on surface
temperatures can explain the deviations in precipitation scaling from CC rates in other tropical regions
too. Furthermore, our methodology to remove the cloud cooling effects on surface temperatures could be
extended to derive scaling relationships of other, observed variables to obtain their response to global
warming as well. Our findings add a novel component to better interpret precipitation scaling rates derived
from observations to support climate model projections.

# Data Availability

The daily gridded precipitation and temperature datasets were obtained from the Indian Meteorological
department (IMD, https://cdsp.imdpune.gov.in/home_gridded_data.php (doi: 10.1029/2008GL035143).
The APHRODITE (Asian Precipitation Highly Resolved Observational Data Integration towards
Evaluation) dataset is available at http://aphrodite.st.hirosaki-u.ac.jp/products.html. Sub-daily
precipitation data at 3 hourly resolution was obtained from TRMM (Tropical Rainfall measuring mission)
TMPA_3B42_V7        data        (doi:        10.5067/TRMM/TMPA/3H/7)
https://disc.gsfc.nasa.gov/datasets/TRMM_3B42_7/summary. Station-based daily precipitation -
temperature data was taken from NOAA – GSOD sites (Station id: 43295099999, 43003099999 and
43279099999)   at   https://www.ncei.noaa.gov/access/search/data-search/global-summary-of-the-day.
Surface and TOA gridded radiative flux datasets are obtained from NASA CERES EBAF data (doi:
https://doi.org/10.5067/Terra-Aqua/CERES/EBAF_L3B.004.1) and NASA CERES Syn1deg data (doi:
10.5067/TERRA+AQUA/CERES/SYN1DEG-1HOUR_L3.004A)   at   https://ceres.larc.nasa.gov/data/.
Daily dew point temperature data is obtained from the ERA-5 reanalysis (doi: 10.24381/cds.e2161bac).

# Acknowledgements

The author thanks the NASA CERES team for making the satellite data openly available (doi:
10.5067/Terra-Aqua/CERES/EBAF_L3B.004.1   and   10.5067/TERRA+AQUA/CERES/SYN1DEG-
1HOUR_L3.004A) and the Copernicus Climate Change Service for the access to the ERA-5 reanalysis
data (doi: 10.24381/cds.e2161bac).



## Author Contribution

All the authors contributed to the idea and development of the hypothesis. SAG carried out the data analysis. The writing of the manuscript was done by SAG with inputs and edits from AK. AK and SG helped in designing the study. All the authors contributed to the interpretation of the results.

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



**Figures:**

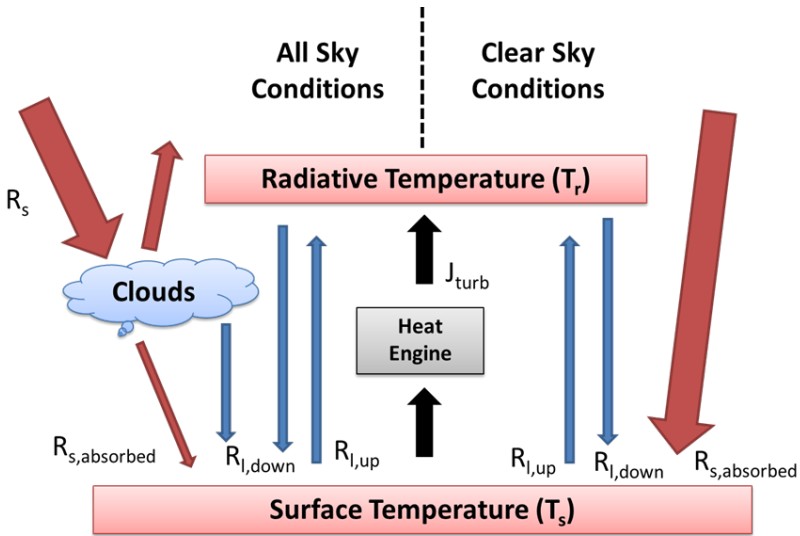

**Figure 1.  Schematic diagram of the surface energy balance, the fluxes of solar (red) and terrestrial**
**(blue) radiation, as well as the turbulent heat fluxes (black).  We consider turbulent heat exchange**
**being driven primarily by an atmospheric heat engine that operates at the thermodynamic limit of**
**maximum power.**



**Figure 2: Comparison of monthly mean temperature time series for observed (IMD) and estimated "all-sky" surface temperatures, averaged over all grid points. (B) Regression between the two temperatures at the grid-point scale. (C) Spatial variation of the root mean squared error (RMSE) in temperature estimates from maximum power compared to observed temperatures.**



41

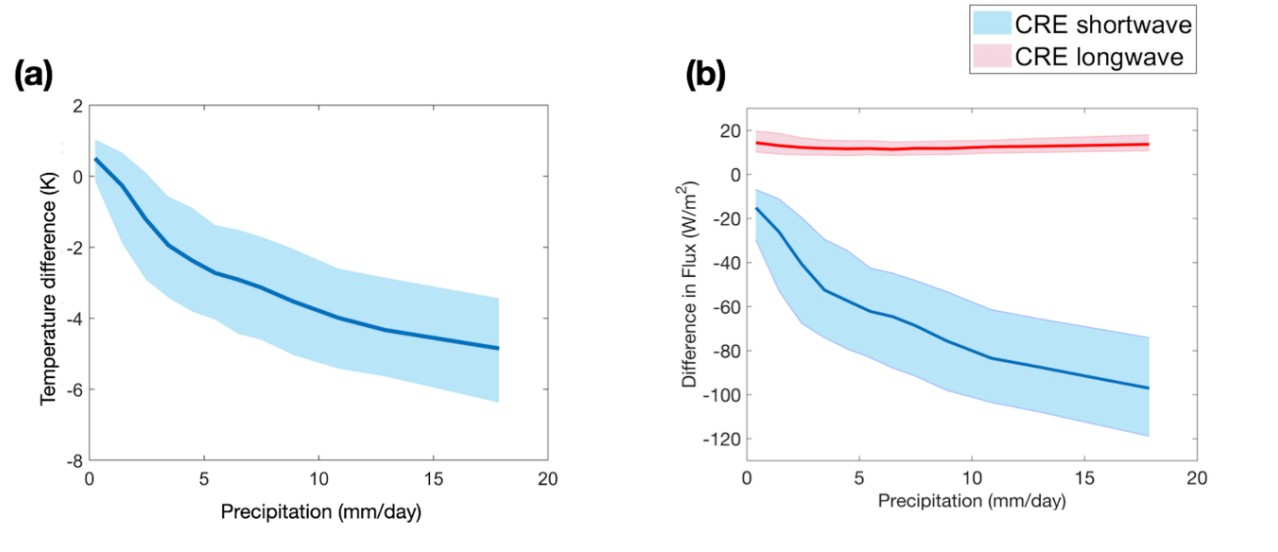

42

**Figure 3: (a) Cooling effect of clouds on surface temperatures calculated from the difference of "all-sky" to "clear-sky" surface temperatures as a function of precipitation over the Indian region. (b) Difference in net shortwave and downwelling longwave radiative fluxes ("Cloud Radiative Effect", CRE) between "all-sky" and "clear-sky" radiative conditions at the surface as a function of precipitation. This was inferred using NASA – CERES (EBAF ed4.1) dataset (Loeb et al., 2018).**

49





**Figure 4. Regional variation of (a) mean daily extreme precipitation (99th percentile) (b) the temperature difference between "clear-sky" and "all-sky" radiative conditions averaged during extreme precipitation events (c) "All-sky" surface temperature during the occurrence of the event (d) Mean number of rainfall days per year**



**Figure 5. (a) Extreme precipitation-temperature scaling using observed (yellow), "all-sky" (red) and "clear-sky" (blue) temperatures over India. (b) Same as (a), but using dew point temperatures. (c) Relationship between dew point temperatures and "all-sky" (red) and "clear-sky" (blue) temperatures. The shaded areas represent the variance in terms of the interquartile range for each bin. Grey dotted lines indicate the Clausius-Clapeyron scaling rate. Note: Logarithmic vertical axis for figure (a,b)**



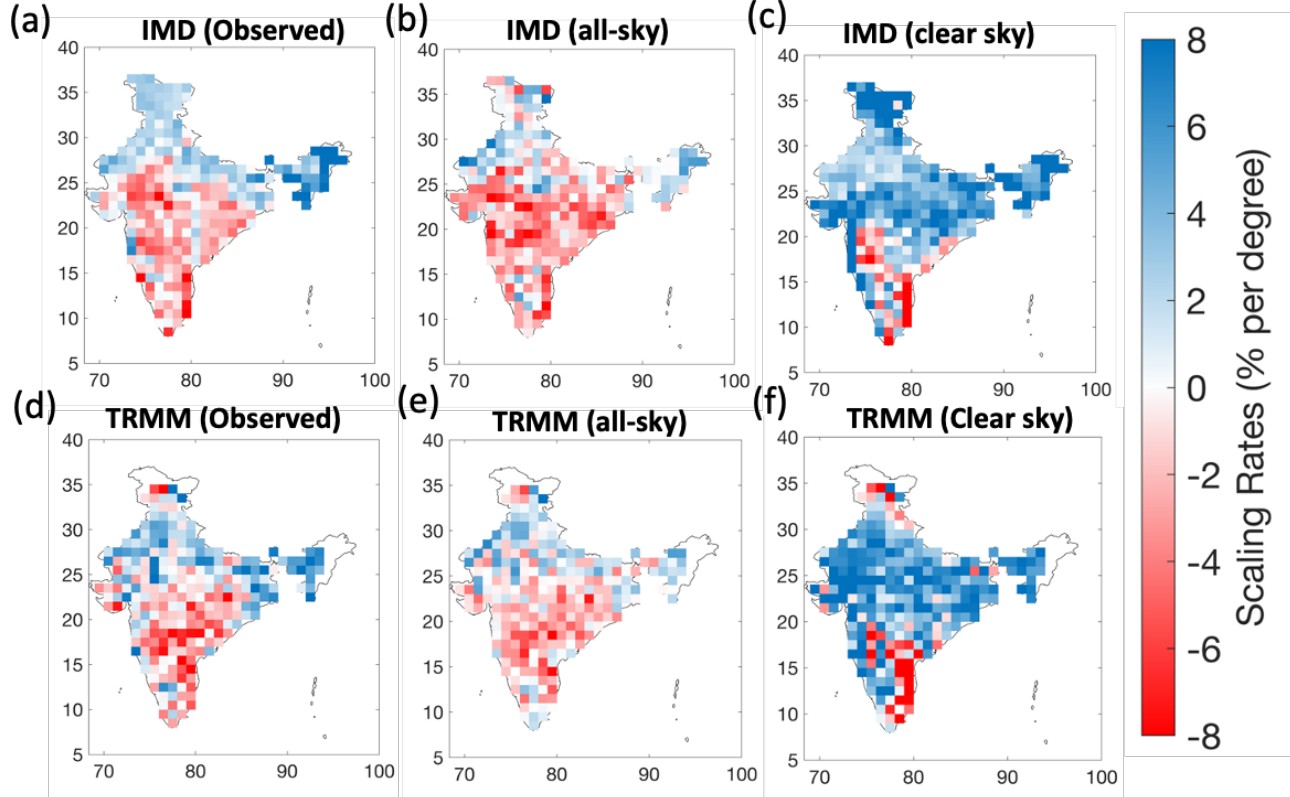

**Figure 6. Regional variation of 99th percentile precipitation-temperature scaling rates using daily (a-c) and 3 hourly (d -f) rainfall data with observed temperatures (a, d), "all-sky" temperatures (b, e) and "clear-sky" temperatures (c, f).**



## Appendix A: Validation of scaling results using station-based GSOD data

We used three station-based daily observations from global surface summary of the day (GSOD) data provided by National Oceanic and Atmospheric Administration (NOAA). We used the data at Mumbai, Chennai and Bangalore Airport to produce the scaling curves (Appendix A). The choice of the station was based to ensure the robustness of results using gauge data as well as to check the effect of seasonality as the three sites receive rainfall during different period of the years. In Mumbai, rainfall occurs mainly during the summer monsoon season while in Chennai heavy rainfall occurs during the winter months (November and December). On other hand, Bangalore receives rainfall during both summer and winter monsoon season (Fig. A1 a – c). Negative scaling was found over these three stations using observed (yellow) and "all-sky" (red) temperatures while with "clear-sky" temperatures (blue), we find positive rates largely consistent with the CC rate (Fig A1 d - i).

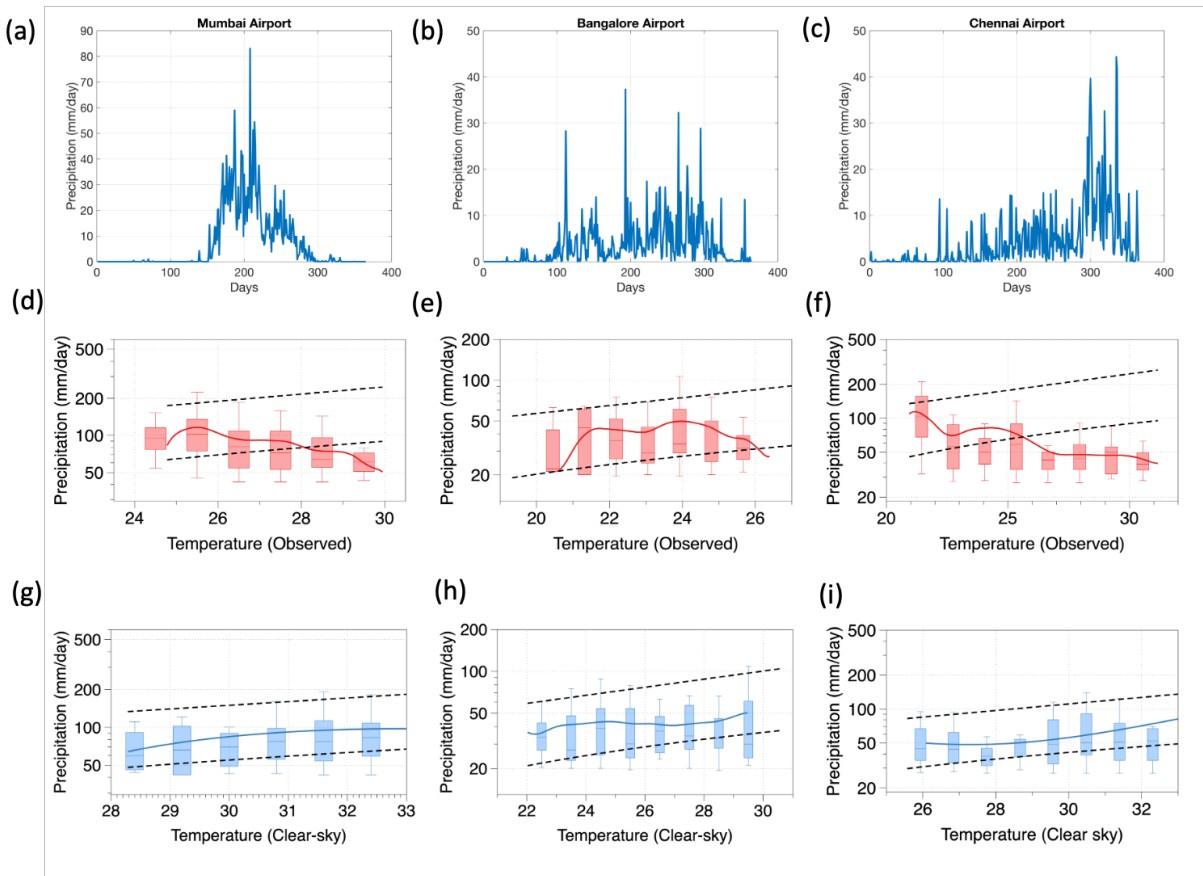

81

**Figure A1. (a-c) shows the annual cycle of mean daily precipitation over GSOD sites in Mumbai airport, Bangalore airport and Chennai airport respectively. Extreme precipitation – temperature scaling curves for (d-f) observed temperatures in red and (g-i) "Clear-sky" temperatures in blue are presented for all the three sites. Red/Blue solid lines indicate the LOESS regression lines. Grey dotted lines indicate the Clausius-Clapeyron scaling rate. Note Logarithmic vertical axis.**





## Appendix B: Validation of scaling results using APHRODITE dataset


Figure B1 shows the spatial variation of daily precipitation – temperature scaling rates estimated from
quantile regression (similar to Fig. 6 in the main text) using the APHRODITE (Asian Precipitation –
Highly Resolved Observational Data Integration towards Evaluation of water resources) dataset (Yatagai
et al., 2012). The results show a diametric change in scaling from being negative for observed and "all-
sky" temperatures to coming close to CC rate (7%/K) for "clear-sky" temperatures. The findings were
consistent with that obtained using the IMD and TRMM dataset (Figure 6).

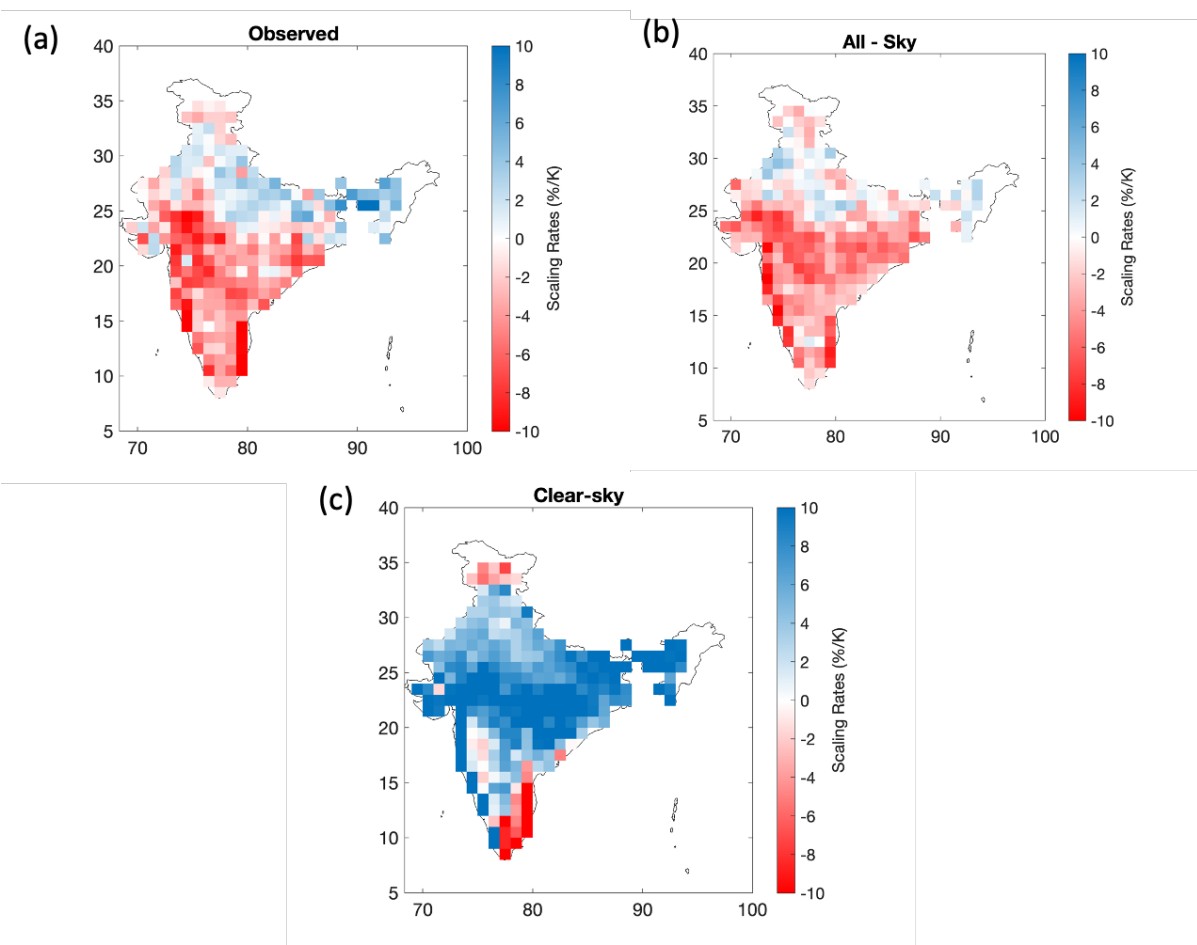


**Figure B1. Regional variation of 99th percentile daily precipitation-temperature scaling rates using (a)**
**Observed (b) "all-sky" and (c) "clear-sky" temperatures. Note: Precipitation data is from APHRODITE**





## Appendix C: Effect of seasonality on scaling rates

To understand the role of seasonality on precipitation – temperature scaling. We divided the precipitation period into two seasonal subsets i.e., summer monsoon season (April to September) and winter monsoon (October to March). Season wise scaling curves (estimated using LOESS regression) are presented in figure C3. We find that observed scaling is uniformly negative in summer over Indian region while during winter the scaling is positive (Fig C3-a, d). This is not surprising because the "hook" or breakdown in scaling happens at high temperature which leads to negative scaling in summer (Figure 5a). Reconstructed "All-sky" temperature showed scaling pattern consistent with observations (Fig. C3- b,e). When scaled with "clear-sky" temperatures, we observed a change in scaling for summer as it turns positive and come close to CC rate. While for winter the scaling remains the same (already positive). It is also important to note that almost 80% of total rainfall over India occurs during the summer monsoon season (Fig C1). As a result, the cooling effect of clouds is mainly experienced during the summer monsoon (where we observed a change in scaling) while the cooling effect remains less than 1K during the winter season (Fig C2). Thus, one does not see a change in scaling between "all-sky" and "clear-sky" conditions for winter season.





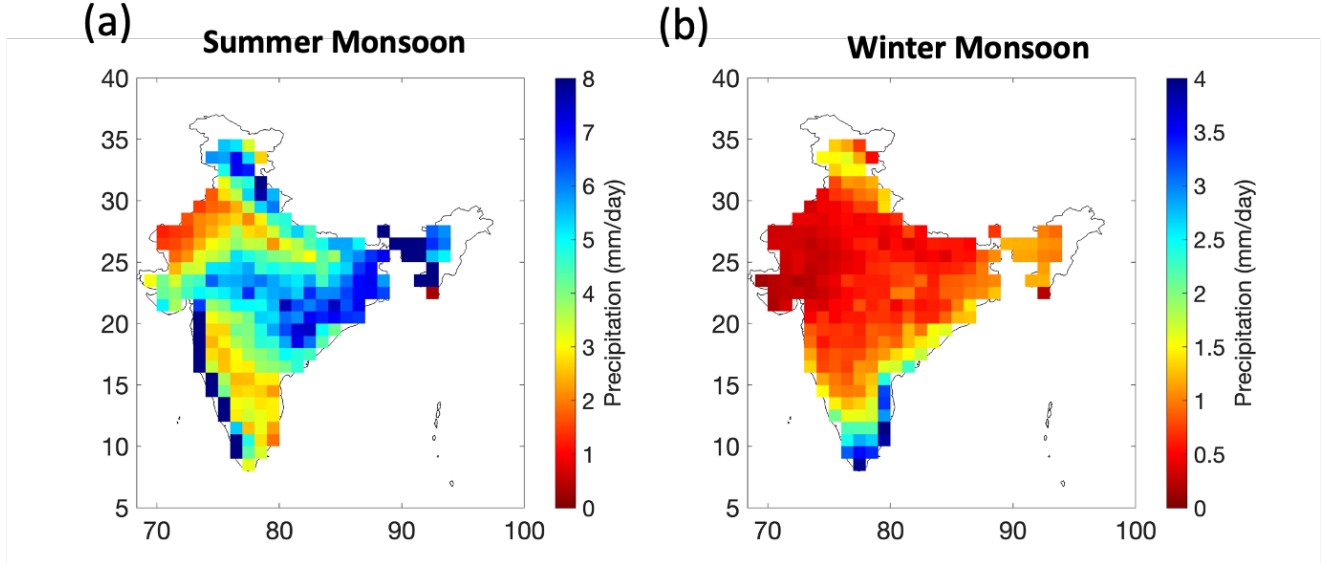

**Figure C1. shows the map of mean daily precipitation during (a) summer monsoon (April – September) and during (b) winter monsoon (October – March).**



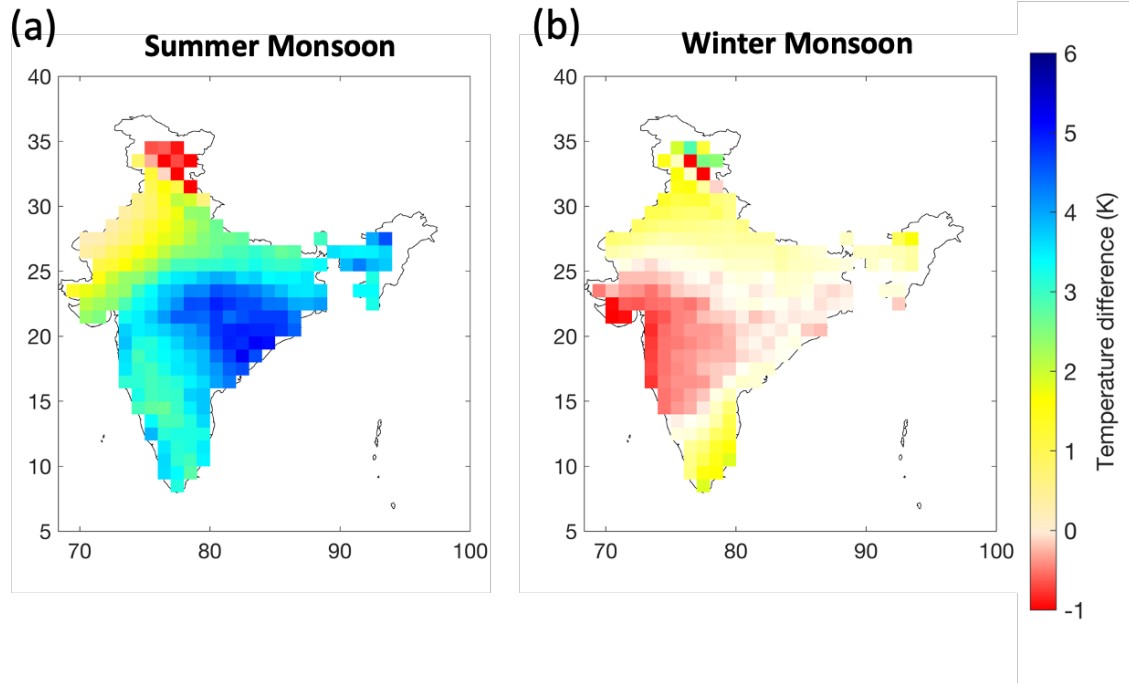

**Figure C2. Shows the map of cooling of surface due to clouds (defined as the difference between "clear-sky" and "all-sky" temperatures) for (a) Summer monsoon (April – September) and (b) Winter monsoon (October – March)**


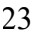

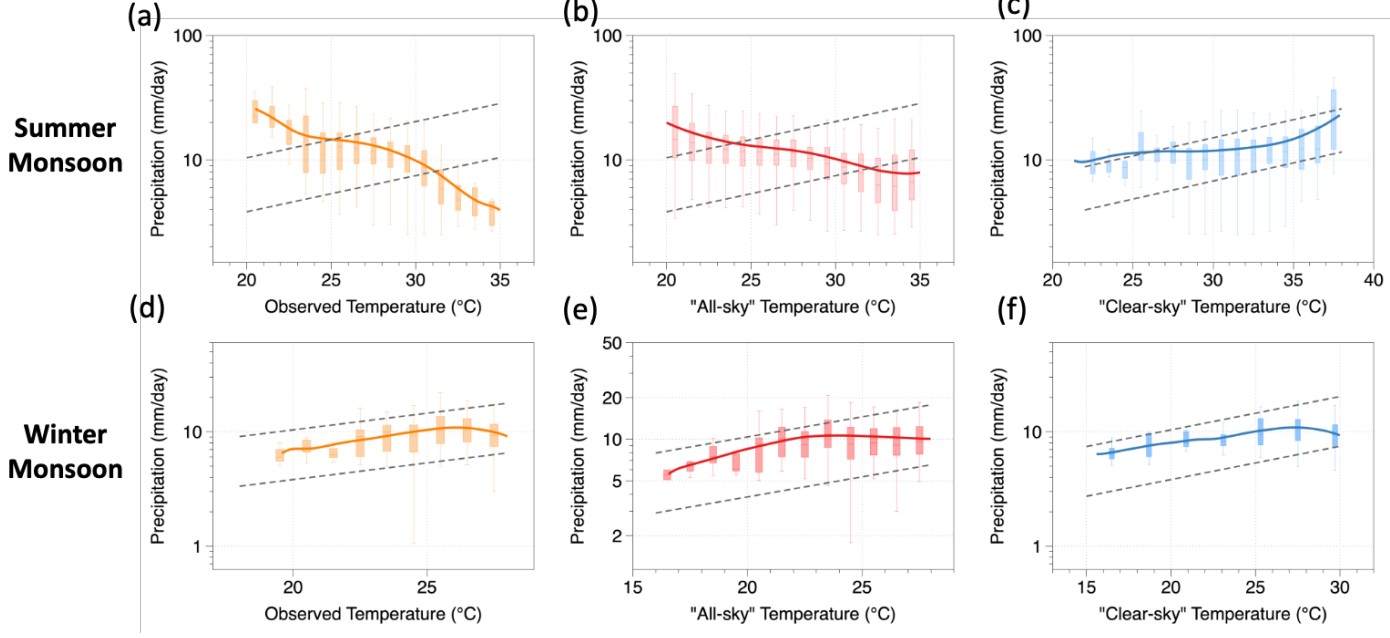

**Figure C3. Extreme precipitation - temperature scaling during summer monsoon (a - c) and winter monsoon (d-f). Scaling curves are shown in orange (a,d) for observed temperatures, in red (b,e) for "all-sky" temperatures and in blue (c,f) for "clear-sky" temperatures. Orange/red/blue solid lines indicate the LOESS regression lines.  Grey dotted lines indicate Clausius – Clapeyron scaling rate. Note: Logarithmic vertical axis. Dataset used is IMD.**