# Peer review of "Break in precipitation – temperature scaling over India predominantly explained by cloud-driven cooling"

_Hydrology and Earth System Sciences, 2022_

## Author Comment (AC1)

We sincerely thank the referee for the encouraging assessment of our work and for highlighting some points which were unclear. In the following we have tried to answer and accordingly incorporate the referee's suggestions. The referee comments are shown in black and our responses are in blue.

**Comment 1:** Lines 232-235, Figure 4b: the differences between the two temperatures are positive in most regions. However, "clear sky" temperature is lower "all sky" temperature in the northernmost part (red). Can authors explain why?

**Response 1:**

We thank referee for pointing this out. The differences in "clear-sky" and "all-sky" temperatures are shaped by changes in the shortwave and longwave radiations due to clouds. Clouds reduces the solar insolation at the surface (hence a cooling effect) but also increases the downwelling longwave radiation (a warming effect). Over most of the Indian region the reduction in shortwave is substantially greater than increase in downwelling longwave (Figure 3b) and thus they produce a localised cooling effect over most regions. However, the Northern most part of India is the high-altitude cold Himalayan region. These high-altitude regions are more sensitive to changes in longwave radiations. As a result, there is a significant increase in longwave radiation with increase in cloud cover which compensates for the cooling due to reduction in shortwave. In the map shown in figure 4b we have isolated extreme precipitation days which corresponds to highly overcast conditions. Due to this compensating effect of longwave radiations, "Clear sky" temperature is lower than "all-sky" temperature over those grid points.

To show this with observations, we have attached a figure which show the mean annual cycles of difference in fluxes (clear sky – all sky) for both shortwave, longwave and total heating over these grid points.

A note about this will be added in the revised version of the manuscript

[Figure]

**Figure shows the daily mean annual cycles of difference in "clear-sky" and "all-sky" fluxes of net shortwave (red), downwelling longwave (blue) and total radiative heating (black) for the grid points over the Himalayan region of India**

**Comment 2:** The modeled monthly temperature is evaluated in section 3.1. However, the daily temperature is used for the precipitation-temperature scaling. How good is the daily temperature calculated from equation (7)?

**Response 2:**
We thank the reviewer for highlighting out this point. The RMSE in the figure 2 was already shown for daily temperatures, However, monthly temperatures were used in timeseries comparison and regression plot which made things unclear and created confusion. We have now modified all the sub-plots in figure 2 for daily temperatures.

**Modified Figure 2:**

[Figure]

**Figure 2: Comparison of daily annual cycle of temperature for observed (IMD) and estimated "all-sky" surface temperatures, averaged over all grid points. (B) Regression between the two temperatures at the grid-point scale. (C) Spatial variation of the root mean squared error (RMSE) in temperature estimates from maximum power compared to observed temperatures.**

**Comment 3:** The radiative effect of cloud on surface temperature can explain the breakdown of the P-T scaling curve. Can authors provide us with the observation-based cloud information to further validate the statement? For example, the map of cloud over.

**Response 3:**

As per referee's suggestion, we have now added the map of observed cloud area fraction from NASA-CERES dataset into the Appendix.

**Figure added:**

[Figure]

Figure: shows the mean cloud area fraction map over Indian region averaged over wet days. Dataset used is NASA-CERES Syn1deg.

**Comment 4:** Extreme precipitation increases monotonically with temperature when the cloud cooling effect is removed. Is there any implication for the future prediction of extreme precipitation based on the monotonical P-T relationship?

**Response 4:**
This is a very important point. The answer is yes, the monotonic increase of precipitation extremes with "clear-sky" temperatures has major implications for future predictions. One of the major issues with peak structure in scaling is that it raises a question if future intensification in precipitation extremes will be constrained by this peak and not increase beyond this point. However, studies using climate model simulation have shown a positive increase in these extremes which is in contrast to estimated negative scaling at high temperatures (See also yin et al., 2021). After resolving the peak by removing the cloud radiative effects, we find a monotonic increase in extremes with temperature which is consistent with model simulations and implies that the peak will not constrain the increase in extremes with anthropogenic warming. Although in this study our main focus was on understanding the "peak" and we have not explicitly used climate model simulations but it remains an important area for future research and a complementary study on this work. A small discussion about the same will also be added in the revised version of the manuscript.

**Minor comments:**

Line numbers are not continuous.
Will be corrected

Line 100: the equation (1) -> equation (1)
Will be corrected

 Line 121: "Detailed derivation about the same can be found in ……". I guess that "same" here is a typo.
Will be corrected

Line 201: following -> aforementioned
Will be changed

Figure 1: the symbols used for the flux-type variables in the figure are not consistent with the symbols in the text.
Will be corrected

Figure 2: in the figure caption, (a) is missing, and (B) and (C) should be lower cases. In addition, 2003 should be placed on the beginning the curve. Otherwise, it gives a wrong impression that the data before 2003 is used in this study.
Will be corrected

Appendix A: observed (yellow) and "all-sky" (red) temperatures are mentioned. However, there is no "all-sky" temperature in Figure A1.

The figure is now modified and the scaling with "all-sky" temperatures is added.
**Modified figure A1:**

[Figure]

**Figure A1. (Row 1)** shows the annual cycle of mean daily precipitation over GSOD sites in Mumbai airport, Bangalore airport and Chennai airport respectively. Extreme precipitation – temperature scaling curves for observed temperatures (yellow), "all-sky" temperatures (red) and "Clear-sky" temperatures (in blue) are presented for all the three sites. Yellow/Red/Blue solid lines indicate the LOESS regression lines. Grey dotted lines indicate the Clausius-Clapeyron scaling rate. Note Logarithmic vertical axis.

Figure A1: it is better to use the actual date instead of days for x-axis in (a), (b), and (c). As such, one can see the seasonal variation of precipitation from the figure more clearly.

As per referee's suggestion, Months are added to the x-axis to better understand the seasonal variations.

---

## Author Comment (AC2)

We sincerely thank the referee for the encouraging assessment of our work and for highlighting some points which were unclear. In the following we have tried to answer and incorporate the feedbacks from the referee. The referee comments are shown in black and our responses are in blue.

**Comment 1:** In Figure 1, the monthly temperature is used despite of the daily scale for analysis. This may need to be fixed.

**Response 1:**
We thank the referee for highlighting out this point. The RMSE in the figure 2 was already shown for daily temperatures, However, monthly temperatures were used in timeseries comparison and regression plot which made things unclear and created confusion. We have now modified all the sub-plots in figure 2 for daily temperatures.

[Figure]

**Figure 2: Comparison of daily annual cycle of temperature for observed (IMD) and estimated "all-sky" surface temperatures, averaged over all grid points. (B) Regression between the two temperatures at the grid-point scale. (C) Spatial variation of the root mean squared error (RMSE) in temperature estimates from maximum power compared to observed temperatures.**

**Comment 2:** There seems to be minor difference between the clear sky scaling in IMD and TRMM in foothill of Himalayas north of India in the figure 6, does the authors know why?

**Response 2:**
We thank referee for pointing this out. There exist differences in the scaling between IMD and TRMM data in the Himalayan region of India. To our understanding, this is mainly because of the under-estimation of rainfall by TRMM over this region which had also been documented by several studies (Kanda et al., 2020; Sharma et al., 2020; Shukla et al., 2019). We will add a note about it in the revised version of the manuscript.

**Comment 3:** On figure 6, while the removal of the effect in the north-west India in IMD (Fig. 6c) is relatively similar compared to the Observed, the TRMM data (Fig. 6e) exhibits more cooling, does the authors know why? Maybe it's because TRMM are more affected by clouds covers?

**Response 3:**
Yes, we see a stronger increase with TRMM data. We agree with referee and we too think that, this is mainly because TRMM is a satellite-based dataset like CERES and thus it may respond more consistently/strongly to cloud radiative effects then compared to IMD data.

**Comment 4:** In the introduction, the authors used the "here we ….", which seems pretty weird, this is in Line 57, 64, 83. It may be better if the authors use the expression such as "In this study" or words of similar sorts.

**Response 4:**
As per referee's suggestion, the text is now accordingly modified for these lines.

---

## Author Comment (AC3)

We sincerely thank the referee for the thoughtful assessment of our work and for highlighting some points which were unclear and needs to be further discussed. In the following we have tried to answer and accordingly incorporate the referee's suggestions. The referee comments are shown in black and our responses are in blue.

**Comment 1:** During the period winter monsoon, the precipitation-temperature scaling has a drop over the western India after removing the radiative effects of clouds (Figure C2). The results may imply that the scaling of extreme precipitation-temperature is not constrained by the radiative effects of cloud. Then, the generalizability of the conclusions revealed by this study may be questionable. Therefore, more explain or discussion about the drop scaling of P-T during the period of winter monsoon are essential.

**Response 1:**
Figure C2 shows the cloud radiative cooling of surface temperatures during summer (a) and winter monsoon (b). Scaling curves were presented for the two seasons in figure C3. While we see a diametric change in scaling for "all-sky" and "clear-sky" temperatures during summer monsoon, the scaling remains largely same during winter with a slight drop at high temperatures. There are two aspects to this question. 1) why the scaling does not change for "all-sky" and "clear-sky" temperatures during winter months? and 2) why we still see a slight drop in scaling during winter?
To answer (1): Winter months are associated with lower solar insolation. Unlike summers, the radiative effect of clouds on shortwave radiation does not substantially exceeds the compensating heating effect by change in longwave radiation. Over India, these months are further characterized by low rainfall and lower cloud cover. As a result, we don't see much difference between the "all-sky" and "clear-sky" conditions and their difference remains close to 1 K (figure C2) and thus the scaling for "all-sky" and "clear-sky" temperatures remains same.
To answer 2) we still see a slight drop in scaling during winter months. To our understanding this drop could be due to moisture availability limitation at high temperature (An effect which had been argued by previous studies (Hardwick et al., 2010). Other factors that can also play a role could be the spread in cloud radiative cooling which largely arises from the inconsistencies between precipitation and radiation datasets or the effect of changing rainfall type which is not explicitly considered in our study.
The discussion about it will be added in the revised version of the manuscript.

**Comment 2:** South of India, the cloud-driven cooling does not play a role in the extreme precipitation-temperature scaling (Figure 6 and B1). This phenomenon needs further explanation by combined other factors (e.g., the topography, distributions of sea and land).

**Response 2:**
Negative scaling was found over some grids in South of India for both "all-sky" and "clear-sky" temperatures. We think that "clear-sky" temperatures could not resolve this negative scaling due to the following reasons:

1) These are the grids which receives contribution from rainfall during both summer and winter monsoon, However, a relatively higher proportion of the rain happens during winter monsoon (Figure C1). The reason being that this region lies over the

leeward side of Western ghats for the incoming southwest monsoon winds during summer monsoon. Whereas during the winter monsoon, Northeast winds blow over Bay of Bengal leading to large moisture advection and more rain over this region. As a result of this seasonality effect more extreme precipitation are sampled during winter season over this region while during the summer season, moisture supply may limit these extremes to increase. This may lead to a negative scaling when a single quantile regression slope is fitted over the whole temperature range.

2) Another reason could be the development of low-pressure system in Bay of Bengal during winter months which causes cyclones over the Eastern coast of India. These cyclonic systems thus cause very high rainfall at low temperatures which causes negative scaling. More work is still needed to be done to resolve these systems in the conventional scaling approach and remains out of scope for present study.

The discussion about it will be added in the revised version of the manuscript.

**Comment 3:** The scaling of precipitation-temperature after removing the cloud cooling effects are not equal CC rate (~7%/°C) strictly, the effects of other factors that could influence the scaling should be discussed in Section 4.

**Response 3:**
We agree with the reviewer that while the spatial aggregation of all grids correspond to a CC scaling of precipitation extremes over Indian regions (Figure 5a), there still exist regional variabilities at grid scale and the scaling rates are not strictly equal to CC rates. We believe that these deviations could be due to the following reasons:

1: Present scaling approach does not explicitly consider the contribution from the large-scale dynamics and regional circulation patterns which can cause local changes in the scaling estimates.
2: Change in rainfall types - Orographic, stratiform or convective can affect the estimates of scaling rates.
3: Inconsistency between precipitation and radiation datasets can also cause uncertainties in estimating the cooling associated with rainfall event and can affect the estimates of scaling rates.

These points will be added in the revised version of the manuscript.

**Comment 4:** The Figure 1 given the temperature after removing the cloud effects at monthly scale. However, this study did not involve the scaling of precipitation-temperature at monthly scale, so the Figure could be replaced by daily data.

**Response 4:**

We thank the reviewer for highlighting out this point. The RMSE in the figure 2 was already shown for daily temperatures, However, monthly temperatures were used in timeseries comparison and regression plot which made things unclear and created confusion. We have now modified all the sub-plots in figure 2 for daily temperatures.

**Modified Figure 2:**

[Figure]

**Figure 2: Comparison of daily annual cycle of temperature for observed (IMD) and estimated "all-sky" surface temperatures, averaged over all grid points. (B) Regression between the two temperatures at the grid-point scale. (C) Spatial variation of the root mean squared error (RMSE) in temperature estimates from maximum power compared to observed temperatures.**

---

## Author Response (AR1)

**Letter to Editor: Submission of revised manuscript**

Dear Editor,

Thank you for providing us the opportunity to revise the manuscript. We thank the three anonymous reviewers for their constructive comments and positive assessment of our work. We have tried to incorporate the feedbacks from the referees and believe that the manuscript is improved after these changes. Most of the referee's suggestions were directed towards adding discussion about the relevant aspects which was missing from the older version which are now added.

In the following, we have summarised the changes made in the manuscript. A point-by-point response to referee's comments was already uploaded as "Responses to referee comments".

**Changes made:**

- Monthly temperatures were used in timeseries comparison and regression plot which made things unclear and created confusion. We have now modified all the sub-plots in figure 2 for daily temperatures. (Commented by all the referees)

**Based on Referee 1 Comments:**
- Discussion is added about the lower "clear-sky" temperatures over Northern Himalayan regions. Lines added in final manuscript: 236 – 240.
- Observation based cloud cover map was added in Appendix C (Figure C1).
- Implication for the future prediction of extreme precipitation based on the monotonical P-T relationship is added in lines: 344-346.
- Symbols for figure 1 are made consistent with symbols used in text
- Figure A1 is modified and "all-sky" scaling (red) is included.
- All other minor comments are corrected.

**Based on Referee 2 Comments:**
- Discussion about the possible causes for the negative scaling obtained with "clear-sky" temperatures over South India is now added. Lines added in final manuscript: 323 – 337.
- Discussion about the role of seasonality is added. Lines added in final manuscript: 338 – 342.
- Discussion is added in section 4 about the factors that could influence the scaling and their deviations from CC rates (7%/K). Lines added in final manuscript: 370 – 377.

**Based on Referee 3 Comments:**
- Discussion is added about the minor difference between the clear sky scaling in IMD and TRMM in foothill of Himalayas north of India. Lines added in final manuscript: 320 – 322.
- Line 57, 64, 83 of the previous manuscript are revised as per referee's suggestion.

We think that we have adequately addressed the concerns raised by the reviewers and hope that you find the manuscript acceptable for publication. As corresponding author, I confirm that manuscript has been read and approved by all the co-authors.

Thank you for your consideration.
Sincerely,

Sarosh Alam Ghausi

Doctoral Researcher

Max Planck Institute for Biogeochemistry

E-mail: sghausi@bgc-jena.mpg.de